# Protocol for a Delphi consensus study to select indicators of high-quality general practice to achieve Quality Equity and Systems Transformation in Primary Health Care (QUEST-PHC) in Australia

**Phyllis Lau** [1,2] *, **Samantha Ryan** [1,2], **Penelope Abbott** [1,2], **Kathy Tannous** [2,3], **Steven Trankle** [1,2], **Kath Peters** [2,4], **Andrew Page** [2], **Natalie Cochrane** [1,2], **Tim Usherwood** [5,6], **Jennifer Reath** [1,2]

1 School of Medicine, Western Sydney University, Sydney, NSW, Australia, 2 Translational Health Research Institute, Western Sydney University, Sydney, NSW, Australia, 3 School of Business, Western Sydney University, Sydney, NSW, Australia, 4 School of Nursing and Midwifery, Western Sydney University, Sydney, NSW, Australia, 5 Faculty of Medicine and Health, The University of Sydney, Sydney, NSW, Australia, 6 George Institute for Global Health, Sydney, NSW, Australia

* Phyllis.lau@westernsydney.edu.au

## Abstract

### Background

High-quality general practice has been demonstrated to provide cost-effective, equitable health care and improve health outcomes. Yet there is currently not a set of agreed comprehensive indicators in Australia. We have developed 79 evidence-based indicators and their corresponding 129 measures of high-quality general practice. This study aims to achieve consensus on relevant and feasible indicators and measures for the Australian context.

### Methods

This Delphi consensus study, approved by WSU Human Research Ethics Committee, consists of three rounds of online survey with general practice experts including general practitioners, practice nurses and primary health network staff. The identified indicators and measures are grouped under an attribute framework aligned with the Quadruple Aim, and further grouped under structures, processes and outcomes according to the Donabedian framework. Participants will rate each indicator and measure for relevance and feasibility, and provide comments and recommendations of additional indicators or measures. In the last round, participants will also be asked their views on the implementation of a quality indicator tool. Each indicator and measure will require ≥70% agreement in both relevance and feasibility to achieve consensus. Aggregated ratings will be statistically analysed for response rates, level of agreement, medians, interquartile ranges and group rankings. Qualitative responses will be analysed thematically using a mixed inductive and deductive approach.

**Data Availability Statement:** No datasets were generated or analysed during the current study. All

relevant data from this study will be made available upon study completion.

**Funding:** This study is funded by the Digital Health Cooperative Research Centre https://www.digitalhealthcrc.com/. The funding body is part of the Project Control Group which oversees the conduct of the study, including the design of the study and collection, analysis, and interpretation of data and the writing of the manuscript.

**Competing interests:** The authors have declared that no competing interests exist.

## Discussion

This protocol will add to the current knowledge of the translation of performance guidelines into quality practice across complex clinical settings and in a variety of different contexts in Australian general practice. The Delphi technique is appropriate to develop consensus between the diverse experts because of its ability to offer anonymity to other participants and minimise bias. Findings will contribute to the design of an assessment tool of high-quality general practice that would enable future primary health care reforms in Australia.

## Introduction

The collection, robust analysis and appropriate use of general practice data are critical to informing continuous quality improvement and ensure high-quality primary health care (PHC). Many countries around the world collect PHC data for quality improvement purposes such as monitoring health and utilisation of health services like the Nivel's Primary Care Database in The Netherlands, [1] and for research purpose such as the Clinical Practice Research Datalink (CPRD) in the UK [2].

Australian general practice care data is routinely collected by primary health networks (PHNs) for quality improvement purposes and more recently through the Practice Incentives Program Quality Improvement (PIP QI) initiative, launched in August 2019 by the Australian Government to improve patient outcomes [3–6]. Practices registered for PIP QI are required to regularly submit data collected against ten specified improvement measures to their local PHNs and commit to implementing continuous quality improvement activities in partnership with the PHNs [4]. The general practice data, known as the PIP Eligible Data Set, collected through this process is then uploaded to a national portal and the Australian Institute of Health and Welfare (AIHW) oversees access by researchers and other interested parties to the deidentified PIP Eligible Data Set [4]. The 31 PHNs were established in 2015 across Australia for supporting PHC to identify and meet local health needs, building PHC workforce capacity and the delivery of high-quality care, and integrating local health services to improve patient experience and better use of health resources [5]. They also collect and collate other general practice data from extraction of de-identified data from individual practice-based patient records [5]. However, there is a lack of consistency across PHNs in data content, variability in the quality of the data collected and also in the quality improvement outcomes achieved through this process [7].

A general practice indicator is "a measurable element of practice performance for which there is evidence or consensus that it can be used to assess the quality, and hence change in the quality, of care provided" [8]. A set of standardized and evidence-based indicators to measure and track high quality clinical performance and outcomes in general practice is necessary, not only for the profession's accountability, but also for identifying population needs and gaps in the quality of care received by patients across Australia [7, 9]. Whilst the Royal Australian College of General Practitioners (RACGP) Standards for General Practices underpin accreditation, they are minimum standards for benchmarking purposes [4, 6]. Previous work to develop PHC quality indicators have focused on specific areas of interest, for instance, the Primary Care Practice Improvement Tool (PC-PIT) which is an organisational performance improvement tool focusing only on systems and processes [10]. Even though the collection of data is key to the evaluation of health care quality and services and clinical decision-making,

there is currently not a set of universally agreed comprehensive high-quality indicators in Australia that would identify, measure and reward high-quality general practice.

In 2020, Western Sydney University, in partnership with PHNs in the western Sydney region, conducted a literature review to identify evidence-based indicators and measures, then assessed these in three workshops with general practitioners (GPs), practice managers, nurses, consumers and PHN staff in the western Sydney region [11]. A suite of 79 evidence-based indicators and their corresponding 129 measures of high-quality general practice was subsequently developed. The measures specifically included outcome measures as these are rarely addressed in frameworks of quality PHC [12, 13]. Key literature was also analysed to identify four attributes of high-quality general practice and construct a suitable framework for the indicators and measures [11]. The attributes are expressed as 'accountabilities': accountability to our patients; professionally accountable; accountability to the community and accountability to society [14, 15]. They align with the elements of the Quadruple Aim which states that effective healthcare improvement must take into account the care of individual patients, the health of populations, health care costs and the wellbeing of health care providers, [16] and is increasingly used to monitor and evaluate primary health system performance in Australia and countries like the UK and US [13, 17, 18]. The indicators and measures identified are further grouped under structures, processes and outcomes of high-quality general practice according to a Donabedian framework, [19, 20] and include some "blue sky" measures considered difficult to currently implement but are nonetheless important.

This study extends the previous work by Western Sydney University [11]. Wider consultations with Australian stakeholders will be conducted using a Delphi consensus study with experts to explore the relevance and feasibility of the identified suite of indicators and measures. Experts will include Australian general practices and PHNs involved in quality improvement initiatives. Consultations have been held with consumers, with regards to key patient-reported measures (PRMs). Aboriginal and Torres Strait Islander health and justice health sectors will also be consulted with regard to relevant indictors unique for those populations. These will be detailed elsewhere.

## Materials and methods

### Aim

The overall aim is to establish consensus with experts to contribute to the development of the first comprehensive, evidence-based, professionally endorsed tool for analysing and reporting across all components of high-quality general practice in Australia.

### Study design

This study will use a survey to achieve consensus across an expert group of general practice and PHN staff. The Delphi technique has been selected due to its flexibility and anonymity provided to participants [21, 22]. The survey will consist of three rounds to obtain opinions on a suite of indicators and measures previously developed by the research team to reach consensus on a core set of relevant and feasible high-quality performance indicators [11].

### Project governance

A Project Control Group has been established with the responsibility for overseeing the conduct of the project. The group consists of representatives from the Digital Health Cooperative Research Centre (CRC), and eight primary health organisations: Brisbane North PHN, Central and Eastern Sydney PHN, Nepean Blue Mountains PHN, North Western Melbourne PHN,

South Western Sydney PHN, Western Sydney and then (WentWest), Western Australia Primary Health Alliance, and Western NSW PHN.

A Steering Committee, that meets more frequently, has also been established to provide strategic direction and advice to the research team on dissemination and collaboration with relevant stakeholder groups. This committee consists of the representatives of the primary health organisations and the RACGP, Australian College of Rural and Remote Medicine (ACRRM), Justice Health NSW (New South Wales) and SA (South Australia) Prison Health Service.

## Setting

The study will be undertaken in four states in Australia across regions of the eight primary health organisations: seven PHNs and one primary health alliance comprising three PHNs which support primary care across a less populous state of Australia. These organisations cover a total area of 2,942,817km$^2$ in metropolitan and rural Australia, and a diverse population of over 9.6 million with over 3,000 general practices. The characteristics of the PHNs, their geographical locations and the populations in their regions are summarised in Table 1.

## Sample size

The study will aim to recruit a minimum of 80 participants. A minimum of 17 participants is the recommended minimum sample size for content validity in Delphi studies involving the selection of healthcare quality indicators [23]. In order for this Delphi study to meet the minimum sample size requirement, we must achieve a minimum of 47% retention rate in rounds 2 and 3.

## Participants and recruitment

Participants will include GPs, practice nurses, practice managers and key PHN staff who are familiar with quality improvement initiatives in the context of Australian general practice. People under 18 years old will be excluded.

A purposive and convenience sampling approach will be used. Each of the eight primary health organisations will assist in recruiting eight to ten general practices in their region and nominate two to three key PHN staff. Practices will be purposively recruited to maximize

**Table 1. Characteristics of the primary health organisations involved.**

| Primary health organisation | | Total area | Geographical location | Population in year | Number of general practices in year |
|---|---|---|---|---|---|
| Western Sydney PHN (WentWest) [34] | | 766 km$^2$ | Metropolitan | >1,000,000 in 2020 | 347 in 2020 |
| Nepean Blue Mountains PHN [35] | | 9,063 km$^2$ | Metropolitan | >380,000 in 2020 | 138 in 2020 |
| South Western Sydney PHN [36] | | 6,186 km$^2$ | Metropolitan | 1,019,985 in 2020 | 429 in 2020 |
| Central and Eastern Sydney PHN [37] | | 626 km$^2$ | Metropolitan | 1,637,740 in 2018 | 608 in 2020 |
| Western NSW (New South Wales) PHN [38] | | 433,379 km$^2$ | Rural | 309,900 in 2020 | 110 in 2020 |
| North Western Melbourne PHN [39] | | 3,212 km$^2$ | Metropolitan | 1,640,000 in 2020 | 564 in 2020 |
| Brisbane North PHN [40] | | 3,901 km$^2$ | Metropolitan | 1,004,747 in 2017 | 341 in 2019 |
| WA (Western Australia) Primary Health Alliance | Perth North PHN [41] | 2,975 km$^2$ | Metropolitan | 1,065,744 in 2016 [42] | 248 in 2019 |
| | Perth South PHN [43] | 5,148 km$^2$ | Metropolitan | 965,997 in 2016 [44] | 250 in 2019 |
| | Country WA PHN [45] | 2,477,561km$^2$ | Rural | 548,185 in 2016 | Unavailable |

diversity in regard to geographic location, practice size, and socio-economic status based on the Socio-Economic Index for Areas (SEIFA). An Invitation Pack containing an invitation letter, project information and consent form will be emailed by the PHN to their nominated staff and recruited practices. Each practice will nominate one to two practice staff to participate in the survey. All survey participants will be anonymised to their PHNs and other participants with allocation of a random identification number. A password protected file will be maintained by the research team with participants' identifying information.

## Criteria for the Delphi participants to consider

A total of 79 indicators with 129 measures that had been developed and finalised by the QUEST PHC team in 2020 [11] will be assessed by participants in the Delphi study. (Table 2) They are grouped under the four attributes of high-quality general practice framework aligned with the four elements of the Quadruple Aim [14–16]. Table 3 outlines the four high-quality general practice attributes, their definitions and alignment with the Quadruple Aim and the number of indicators and measures identified under each attribute.

**Table 2. Indicators and measures for assessment by participants.**

| Indicators | Related measures |
|---|---|
| **ATTRIBUTE 1: ACCOUNTABLE TO OUR PATIENTS** | |
| **PERSON CENTRED CARE AND PATIENT-TEAM RELATIONSHIP** | |
| S1: Availability of information for patients | Written and electronic information in appropriate languages |
| P2: Patient input/feedback on health care delivery | Evidence of formal process to consider patient input and incorporate into practice care delivery |
| O3: Patient perceptions of care | Results of PREMs |
| *O4: Patient activation* | *PAM® scores* |
| *O5: Strength of team- patient relationship* | *Results from using validated survey tool* |
| **EVIDENCE-BASED COMPREHENSIVE CARE: PREVENTIVE HEALTH CARE** | |
| P6: Risk factors recorded | % active patients ≥15 years with a BMI recorded who have weight classification (obese, overweight, healthy, underweight) in previous 12 months |
| | % active patients ≤ 15 years with height/length and weight recorded in previous 12 months |
| | % active patients ≥15 years with a smoking status recorded/ updated (current, ex-smoker, never smoked) in previous 24 months |
| | % active patients ≥15 years with alcohol consumption status recorded in previous 24 months |
| | % active patients aged 14–19 years with other substance use recorded |
| | % active patients ≥18 years with BP recorded in previous 24 months |
| *P7: Childhood adverse experiences recorded (blue sky)* | *% active patients aged 0–19 years screened for adverse childhood experiences in previous 12 months (blue sky)* |
| P8: Early detection of cancer | % active patients aged 50–74 years with FOBT recorded in previous 24 months |
| | % active female patients aged 25–74 years without hysterectomy with up-to-date cervical screening |
| | *% active female patients aged 50–74 years with no history of breast cancer screened with mammogram in previous 24 months (blue sky)* |

*(Continued)*

**Table 2.** (Continued)

| Indicators | Related measures |
|---|---|
| P9: Adult vaccination | % active patients ≥65 years immunised against influenza in previous 15 months |
| | % active patients with DM immunised against influenza in previous 15 months |
| | % active patients with COPD ≥15 years immunised against influenza in previous 15 months |
| | % active patients ≥70 years with one dose of pneumococcal immunisation recorded and for Aboriginal and Torres Strait Islander patients ≥50 years two doses at 5-year interval |
| | % active patients >70–79 years with shingles vaccination |
| P10: Childhood vaccination | % active patients ≥4 years who are fully immunised according to guidelines |
| P11: Aboriginal and Torres Strait Islander preventive health care | % active patients identified as Aboriginal and/or Torres Strait Islander with Aboriginal Health Check in previous 15 months |
| O12: Patient perceptions of preventive health discussion | PREMs to include patient report of discussion regarding the following health behaviours/risk factors: healthy eating, exercise/physical activity, risks of smoking/QUIT support, alcohol use, unintentional injuries (home risk factors), unsafe sexual practices, unmanaged psychosocial stress |
| **EVIDENCE-BASED COMPREHENSIVE CARE: CHRONIC CARE** | |
| S13: Systems for management of chronic disease | Use of patient chronic disease registers |
| P14: Systems for management of chronic disease | Use of registers for patient follow up and recall |
| S15: Diabetes: known prevalence | % of active patients with diabetes coded in patient records |
| P16 Diabetes: monitoring CV risk | % active patients with DM and have their BP recorded in previous 6 months |
| | % active patients with DM and have their BMI recorded |
| | % active patients with T2DM and have their total Cholesterol, HDL, triglyceride and LDL levels recorded |
| P17: Diabetes: monitoring renal function | % active patients with DM and have their eGFR (estimated glomerular filtration rate) recorded in previous 12 months |
| | % active patients with DM and have their urine ACR recorded in previous 12 months |
| P18: Diabetes: managing risk | % active patients >60 years with T2DM prescribed a statin |
| P19: Diabetes care: managing complications | % active patients with DM and have their retinal screening performed in previous 24 months |
| | % active patients with DM and have their diabetic foot assessment in previous 12 months |
| P20: Diabetes: monitoring blood sugar control | % active patients with DM and have their HbA1c recorded in previous 12 months |
| O21: Diabetes: optimal management | % active patients with T2DM with HbA1c ≤8% |
| | % active patients with T2DM with BP <140/90 mmHg |
| O22: Diabetes: optimal risk management | % active patients with T2DM with lipids to target in previous 12 months |
| | % active patients with T2DM with microalbuminuria on ACE inhibitor or ARB |
| | % active patients >16 years with DM and not smoking |
| S23: Respiratory disease: known prevalence | % active patients with COPD coded in patient records |
| | % active patients with asthma coded in patient records |
| P24: Respiratory disease: use of spirometry record | % active patients with COPD and have spirometry |
| | % active patients with asthma and have their spirometry recorded in previous 24 months |

(*Continued*)

**Table 2.** (Continued)

| Indicators | Related measures |
|---|---|
| P25: Respiratory disease: monitoring risk factors | % active patients with COPD and have their smoking status recorded |
| | % active patients >15 years with asthma and have their smoking status recorded |
| *P26: Respiratory disease: planning care (blue sky)* | *% active patients with asthma with an asthma management plan (blue sky)* |
| *P27: Respiratory disease: Control (blue sky)* | *% active patients with COPD and have their COPD Assessment Test score recorded (blue sky)* |
| | *% active patients with asthma and have Asthma Control Questionnaire recorded (blue sky)* |
| P28: Respiratory disease: appropriate use of medication | % active patients with COPD on LAMA |
| | *% active patients ≥12 years with asthma on ICS containing preventer (blue sky)* |
| *O29: Respiratory disease: COPD control (blue sky)* | *% active patients with COPD and have been hospitalised in previous 6 months (blue sky)* |
| S30: Cardiovascular disease: known prevalence | % active patients with CVD by category coded in patient records |
| P31: Cardiovascular disease: monitoring CVD risk | % active patients aged 45–74 years with the necessary risk factors assessed (smoking, diabetes, BP, Total Chol, HDL, age, gender) to enable CVD assessment in previous 24 months |
| | % active patients aged 45–75 years with no known CVD and with absolute CVD risk calculated in previous 24 months |
| | % active Aboriginal and/or Torres Strait Islander patients aged 35–75 years with no known CVD and with absolute CVD risk calculated in previous 24 months |
| P32: Cardiovascular disease: monitoring CVD | % active patients ≥18 years with hypertension and have BP recorded in the previous 6 months |
| P33: Cardiovascular disease: management of CVD | % active patients ≥18 years with CVD and have statin prescribed |
| 34: Cardiovascular disease: Optimal outcome | % active patients with hypertension whose most recent BP is <140/90 mmHg |
| 35: Renal disease: known prevalence | % active patients with renal disease coded in patient records |
| P36: Renal disease: screening for renal disease | % active patients with DM screened for nephropathy (eGFR and ACR) in previous 12 months |
| | % active patients coded in patient record as having hypertension screened for nephropathy (eGFR and ACR) in previous 12 months |
| | % active Aboriginal and/or Torres Strait Islander patients >30 years screened for nephropathy (eGFR and ACR) in previous 24 months |
| P37: Renal disease: monitoring renal disease | % active patients with renal disease and had their BP recorded in previous 12 months |
| | % active patients with renal disease and had their eGFR recorded in previous 12 months |
| | % active patients with renal disease and had their urine ACR recorded in previous 12 months |
| | % active patients with renal disease and had their chronic kidney disease stage recorded |
| O38: Renal disease: dialysis | % active patients with renal disease on dialysis |
| S39: Mental health: known prevalence of mental health conditions | % active patients with mental health conditions within each mental health category |
| S40: Mental health: known prevalence of co-morbidity | % active patients with mental health and also diagnosed with each of following: diabetes, CVD, respiratory and renal disease |
| P41: Mental health: treatment planning | % active patients with mental health with a GP Mental Health Treatment Plan (such as MBS item number 2715) in previous 12 months |

(*Continued*)

**Table 2.** (Continued)

| Indicators | Related measures |
|---|---|
| P42: Mental health: management of patients with a mental health diagnosis documented | % active patients ≥15 years with a BMI recorded who have weight classification (obese, overweight, healthy, underweight) in previous 12 months |
| | % active patients ≥15 years with a smoking status recorded/ updated (current, ex-smoker, never smoked) in previous 24 months |
| | % active patients ≥15 years with alcohol consumption status recorded in previous 24 months |
| | *% active patients with follow-up GP visit within 7–30 days of hospital discharge related to psychiatric condition (blue sky)* |
| S43: *Advance care planning (blue sky)* | *% active patients ≥75 years with discussions about advance care planning recorded on file (blue sky)* |
| P44: *Advance care planning (blue sky)* | *% active patients ≥75 years with Advance Care Plan uploaded to My Health Record (blue sky)* |
| **ACUTE CARE: PRESCRIBING SAFETY** | |
| S45: Safe prescribing of opioids and benzodiazepines | Practice has a policy on the safe prescription of opioids and BZDs |
| S46: Safe prescribing of opioids and benzodiazepines | Practice has a policy on discussing safe prescription of opioids and BZDs with all new prescribers |
| O47: Safe prescribing of opioids and benzodiazepines | % acute patients prescribed opioids who had discussion of risk of opioid use with prescriber |
| **ATTRIBUTE 2: PROFESSIONALLY ACCOUNTABLE** | |
| **MULTIDISCIPLINARY TEAM-BASED CONTINUING CARE THAT IS COORDINATED AND INTEGRATED WITH OTHER SERVICES AND THE MEDICAL NEIGHBOURHOOD** | |
| S48: Practice goal/mission | Defined practice mission/goal |
| | Mission/goal accessible to staff |
| | Mission/goal accessible to patients |
| S49: Practice profile | Total number of staff in each professional category including FTE |
| S50: Data sharing with local hospitals | Able to receive electronic discharge summary |
| | *Able to receive data in real time e.g. shared EHR or real time electronic shared care plan (blue sky)* |
| S51: Data sharing with other health care providers | *Practice has GP system for notification of specialist and allied health care correspondence ((blue sky)* |
| S52: Use of My Health Record | % of active patients with Shared Health summaries uploaded to My Health Record |
| P53: Team-based care | Regular clinical review meetings involving all team members |
| | *Assigned care teams to coordinate care for individual patients (blue sky)* |
| | Reports from each team member in patient file |
| P54: Care planning | % active patients with chronic disease who had a GP management plan in previous 12 months |
| | % active patients with chronic disease who had a medication management review (HMR) in previous 12 months |
| O55: GP and staff satisfaction | Survey measuring GP and staff satisfaction with: enjoyment of work, impact on local community health, safety in work, income from work, time with patients, work/life balance |
| O56: *Patient experience of continuity of care (blue sky)* | *PREM questions on time taken for the notification of abnormal test results (blue sky)* |
| O57: Care plan engages patient | PREM questions on experience with care planning |
| | PAM® scores |

*(Continued)*

**Table 2.** (Continued)

| Indicators | Related measures |
|---|---|
| O58: *Follow-up following hospital attendance (blue sky)* | *% of active patients reviewed following ED presentation within 7 days (blue sky)* |
| | *% of active patients reviewed following admission within 3 days (blue sky)* |
| **CLINICAL GOVERNANCE** | |
| S59: Clinical governance systems in place | Practice currently accredited according to RACGP or ACRRM standards |
| **STAFF TRAINING** | |
| P60: Regular staff education undertaken | Number of meetings/attendances recorded |
| P61: Assessment of learning needs | Evidence of process for assessment of learning needs |
| **DATA-ENABLED PRACTICE QUALITY IMPROVEMENT** | |
| S62: Data quality and completeness of demographic and key health data | % active patients with date of birth recorded |
| | % active patients with gender recorded |
| | % active patients with allergy or 'nil known allergy' coded in patient records |
| P63: Improving the quality of our practice | Evidence of work on data cleansing |
| | Data reports and date of most recent report |
| | Evidence of formal review of the collected data |
| O64: *Consumer satisfaction with quality (blue sky)* | *Analysis of validated survey responses (blue sky)* |
| **EDUCATION, TRAINING AND RESEARCH TO SUPPORT QUALITY AND SUSTAINABILITY** | |
| S65: Registered for postgraduate GP training | Accredited as training practice with local RTO |
| P66: Engagement with student training | Number of medical, nursing and allied health students undertaking placements in previous 12 months |
| P67: *Research activity (blue sky)* | *Evidence of engagement with research or PDSA activities (blue sky)* |
| **ATTRIBUTE 3: ACCOUNTABLE TO THE COMMUNITY** | |
| S68: Urgent access to care | Provides same day appointments |
| S69: Access to non-face-to-face care e.g. telephone, email | Process documented and advertised to patients for phone/email access |
| S70: Patient demographics recorded | % active patients with cultural and linguistic status recorded |
| | % active patients who identify as Aboriginal and/or Torres Strait Islander |
| | % active patients with Aboriginal and/or Torres Strait Islander status coded in patient records |
| | % active patients ≥16 years with Australian Government health care card |
| S71: Meets the needs of Aboriginal and/or Torres Strait Islander patients | Practice registered for PIP Indigenous Health Incentive |
| S72: Health related social needs assessed | *% active patients with screening for health-related social needs recorded (blue sky)* |
| S73: Community engagement | Practice has community/patient advisory structures |
| P74: Provides health care to vulnerable communities | Bulk billing for Australian Government health care card holders |
| P75: Meets the needs of CALD communities | Provides bilingual services as required |
| O76: Access to regular primary care provider (as measured in response to PREMs) | *% active patients reporting they have a specific GP/ practice nurse/ care team (blue sky)* |
| | % active patients reporting difficulties obtaining care in previous 12 months |
| | % active patients reporting same day response to phone call to GP/ nurse |

(*Continued*)

**Table 2.** (Continued)

| Indicators | Related measures |
|---|---|
| O77: Access for low SES | *Compare % active patients who are Australian Government health care card holders with % holding Australian Government health care cards in practice LGA (blue sky)* |
| **ATTRIBUTE 4: ACCOUNTABLE TO SOCIETY** | |
| O78: *Avoidable hospital care (blue sky)* | *Use of linked data to measure potentially preventable hospital admissions (blue sky)* |
| O79: *Duplication of care (blue sky)* | *Use of linked data to measure duplication of pathology and radiology services (blue sky)* |

**S** = structural indicators measuring organisation factors that define the health system including material resources (e.g. facilities, equipment, money), human resources (e.g. number and qualifications of staff) and organisation structure (e.g. staff organisation, methods of reimbursement); **P** = process indicators measuring what is actually done in giving and receiving care and can also be thought of as activities; **O** = outcome indicators measuring the effect of care on populations and patients.

## Survey format

Three rounds of online surveys will be administered using the Qualtrics platform. (Qualtrics, Provo, UT, USA. https://www.qualtrics.com). The online survey has been constructed and pilot-tested for comprehension and adequate functioning of the survey set up. Unique links to each round will be emailed to participants on the morning that it is officially opened. Each round will take around 20 to 30 minutes to complete and will remain opened for three weeks. Results will be analysed at least two weeks in between rounds. Participants will receive up to three email reminders to complete each round before it closes.

## Rating process

Participants will be asked to rate each indicator and measure for relevance and feasibility in three rounds of the online survey. Relevance is defined as the value and appropriateness of an indicator/measure in Australian general practice. Participants will be asked to rate on a

**Table 3. High-quality general practice attribute framework, their alignment with the quadruple aim and the number of indicators and measures under each attribute.**

| Attribute | Definition [11] | Aligning with Quadruple Aim [16] | Number of indicators and measures |
|---|---|---|---|
| Attribute One: Accountability to our patients | High-quality general practice provides evidence-based, person-centred and comprehensive care (including preventive, chronic and acute care), with patient-general practice team partnerships as a key aim. | Improving the individual experience of care | 47 indicators with 79 measures |
| Attribute Two: Professionally accountable | High-quality general practice is:<br>■ high-functioning multidisciplinary teams engage in continuing care that is coordinated and integrated with other services and the medical neighbourhood;<br>■ supported by clinical governance, staff training and data-enabled practice quality improvement;<br>■ engaged with general practice education and/or research to provide a means of sustaining the quality of the health system. | Improving the work life of clinicians and staff | 19 indicators with 31 measures |
| Attribute Three: Accountable to the community | High-quality general practice is accessible, responsive to population health needs and focussed on providing equitable care. | Improving the health of populations | 10 indicators with 16 measures |
| Attribute Four: Accountable to society | High-quality general practice promotes efficient stewardship of health resources. | Reducing the per capita costs of care for populations | 2 indicators with 2 measures |

4-point Likert scale: 1 irrelevant; 2 somewhat irrelevant; 3 somewhat relevant; 4 relevant. Feasibility is defined as the applicability and implementability of an indicator/measure in Australian general practice. Participants will be asked to rate each indicator/measure on a 4-point Likert scale– 1 infeasible; 2 somewhat infeasible; 3 somewhat feasible; 4 feasible. Text boxes will be available for participants to provide comments, including recommendations for additional indicators or measures, for each subgroup of indicators.

The flow of the Delphi study rating process is shown in Fig 1. In Round 1, participants will initially be asked to provide demographic information including their name, age, gender, job position, and number of years of experience. They will then be asked to rate the indicators and measures under Attribute 1. In subsequent rounds, only names will be requested to match participants' responses in the various rounds. In Round 2, they will be presented with items that did not reach consensus in Round 1, and given the opportunity to change their previous responses if they wish to do so. They will then be asked to rate the indicators and measures under Attributes 2, 3 and 4 and to provide comments as before for each subgroup of indicators. In Round 3, they will similarly be presented with items that did not reach consensus in Round 2, and given the opportunity to change their previous responses if they wish to do so. In this last round, as the final list of indicators and measures emerges, participants will be presented with a summary of any suggestions or qualitative responses from rounds 1 and 2, and

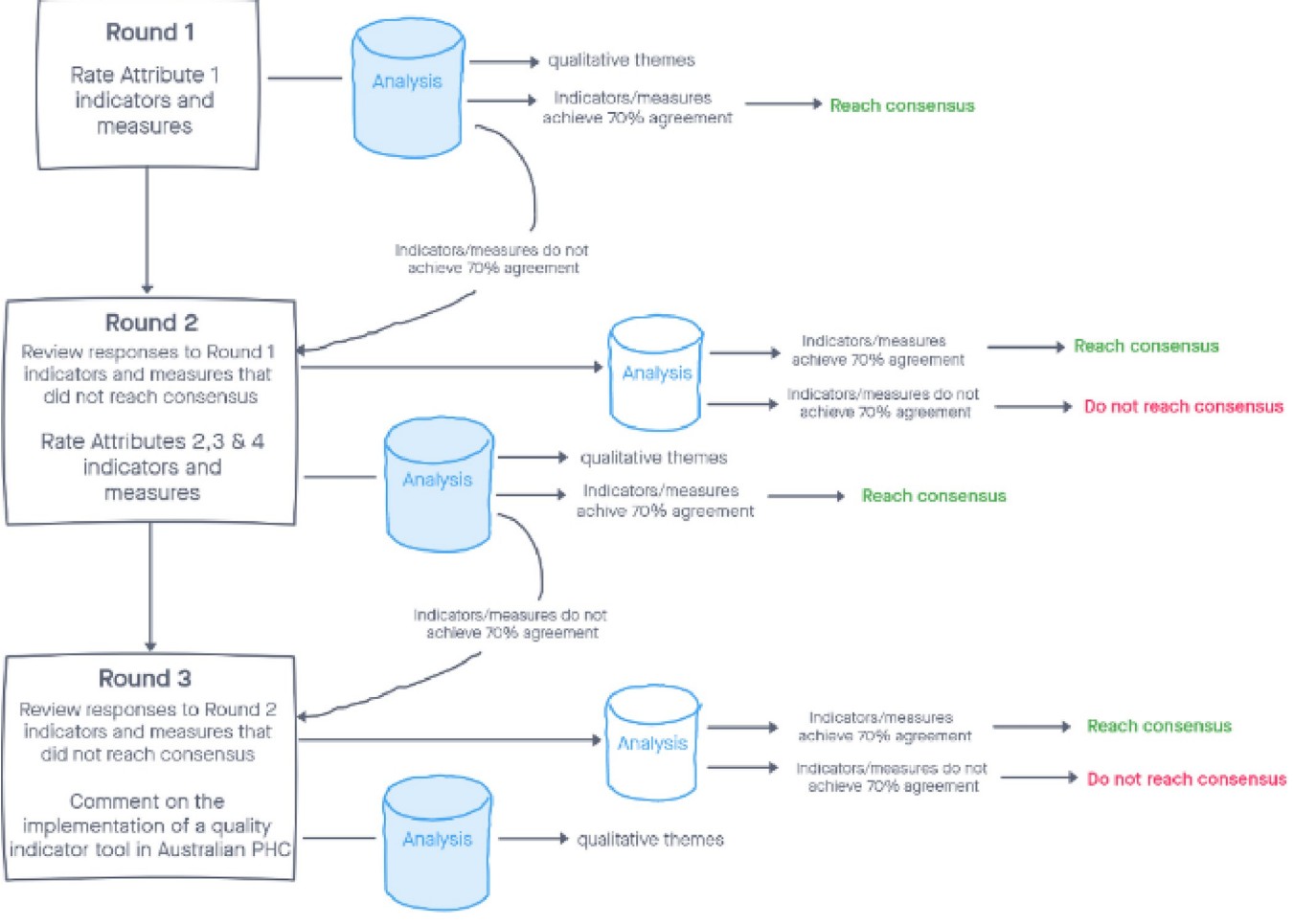

**Fig 1. Delphi survey rating process.**

asked open questions regarding their views and suggestions on the implementation of a quality indicator tool in Australian general practice.

The levels of consensus in the Delphi methodology vary depending on size of the expert panel and the aim of the research [24, 25]. Consensus target for this study are defined 'a priori' based on previous research experience [26]. Each indicator (average score of its measures) and measure will require a minimum of 70% agreement (combined scores of 3 and 4) in both relevance and feasibility to achieve consensus. We determined that this threshold target and approach to be pragmatic and reasonable for establishing consensus across diverse and complex general practice settings.

## Data analysis

**Quantitative data.** Participants' demographics will be analysed descriptively using Microsoft Excel software. The aggregate results of the participants' responses will be analysed for percentage response rates, percentages for each level of agreement for each measure, medians, interquartile ranges and their associated group rankings [27].

A measure will require at least 70% in both relevance and feasibility to achieve consensus. Score of 1 and 2 will be collapsed as irrelevant or infeasible, and scores of 3 and 4 will be collapsed as relevant or feasible. If an indicator or measure achieves ≥70% in relevance but not feasibility, it will be included in a 'blue skies' category for future consideration. If an indicator or measure achieves ≥70% in feasibility but not in relevance, it will be discarded. Sub-analysis of the individual scores 1, 2, 3 and 4 will also be conducted to help us understand better the strength of the consensus.

**Qualitative data.** Participants' responses in the text boxes will be analysed thematically. They will be imported into the NVivo analysis software and coded using a mix of inductive and deductive approaches [28, 29]. Patterns will then be identified from the codes and grouped according to the accountability attributes (deductive approach) as well as to elicit new themes (inductive approach). The research team will separately and collectively analyse the data and resolve any differences in interpretation.

## Data management plans

The types of data that will be produced include demographic data collected on participant consent forms in MS Word/PDF format and electronic survey data. A MS Excel spreadsheet will be created in which participant names will be assigned a number. Participant numbers will be used in place of participant names in naming participant data files for the duration of the project. Survey files will be named using the participant's number, the survey number and the date e.g. Participant1_survery1_190521.

Digital data will be stored on the Western Sydney University's OneDrive system. PL is the administrator and the only person able to provide access to other team members. The only team members with access are PL, SR and JR.

Non-digital data, if any, will be scanned and stored with the digital data. The original hardcopy documents will be stored in a locked filing cabinet in a locked office at Western Sydney University Campbelltown Campus.

All research data and primary materials will be stored for 15 years and then destroyed in accordance to Western Sydney University protocols.

## Potential risks and risk management

Potential risks related to this project include those that may be internal or external. Survey participants may feel inconvenienced by the process required in the study. This includes being

required to read the project information, sign the consent form and complete three rounds of survey. To manage these risks the project aims and purpose will be clearly explained to the participants who are experts familiar with Australian general practice quality improvement initiatives. The study will be of inherent interest to them. It will also be styled to allow easy completion and participants will be able to save their responses and return to them later.

Some participants may be concerned about confidentiality. Although participants will be asked to provide demographic information, their identity and information will be blinded to other survey participants and the PHNs. As detailed above, they will be provided anonymity with a random participant project number that will only be able to be linked to their identifiable information by the research team.

External to the project, risks include the current COVID-19 pandemic and government restrictions on movements. These restrictions and the workload of vaccine roll-out may potentially affect recruitment and participation as PHNs and general practices are directly involved in pandemic prevention and control. The recruitment and survey timeline will be flexible to accommodate any unforeseen interruptions. Each round may also be opened for a longer period if necessary.

## Ethical considerations

This research has ethics approval from Western Sydney University Human Research Ethics Committee (ID H14460). Participants will be required to provide written consent before round 1 of the survey.

## Status and timeline

At the time of manuscript submission, the research has just commenced recruitment of participants. Tentative timeline is outlined in Table 4.

## Discussion

This study protocol describes the research design for a Delphi study to obtain opinions and reach consensus from experts on a core set of relevant and feasible high-quality performance indicators and measures from a suite of indicators and measures previously developed by the researchers in partnership with PHNs in Western Sydney [11]. This protocol will add to the current knowledge of the translation of performance guidelines into quality practice and how best to measure and promote high quality in Australian general practice.

Whilst many PHNs work with general practices to collect data for quality assurance purposes, there is no agreed comprehensive tool that could identify, measure and reward high-quality general practice. Some work has been done in PHNs supporting Patient-Centred Medical Home model of care, but the indicators that were used revolved around processes and system requirement for a team-based approach to deliver this model of care [30, 31]. Although very useful, these indicators are specific to the PCMH models and are dependent on the continuation of funding and evolving policies to support this model of care. Australian general

**Table 4. Project tentative timeline.**

| 26th October to 25th November 2021 | Recruitment |
|---|---|
| 26th November to 17th December 2021 | Round 1 |
| *18th December 2021 to 13th January 2022* | *Holiday seasons break* |
| 14th January to 4th February 2022 | Round 2 |
| 18th February to 11th March 2022 | Round 3 |

practice requires practical and evidence-based indicators and measures of high quality if funding models move to incorporate payment for quality in addition to current throughput payment [32]. Findings from this Delphi consensus study will address the gaps in the literature around establishing consensus on high-quality structural, process and outcome indicators and measures for use across diverse and complex general practice settings, and contribute to the design of an assessment tool that would change how high-quality general practice can be measured and enable future PHC reforms in Australia.

The suite of 79 indicators and their corresponding 129 measures to be evaluated in this Delphi consensus study were derived from robust interrogation of existing literature and extensive consultations with key stakeholders [11]. They are focused on structures, processes and outcomes of care. This Delphi study will enable consideration of their relevance and feasibility within different general practice clinical settings where multi-morbidities and complex interventions are common and the constraints of providing health services are unique. Opinions from our participants will inform and guide the implementation of the developed tool in the real world.

There is growing interest in the processes required to establish assessment tools to identify high-quality health care and service performance. The Delphi technique is appropriate to develop consensus between the diverse stakeholders and experts in the Australian general practice setting because of its flexibility and ability to offer anonymity to participants. It has the benefit of being able to minimise bias from dominant experts compared to other consensus development methods. It provides a platform to canvass suggestions and opinions on implementation of the tool to measure improvement in individual practices and considerations required for specific contexts including cultural and socio-economic factors that may impact achievement of quality indicators. Additionally, the provision of opportunities for participants to review results from previous rounds and to revise their responses is a unique characteristic of the Delphi technique to enable the determination of consensus. A disadvantage of the Delphi technique, however, is that it does not involve direct interactions with the participants and may limit their ability to generate ideas during the consensus process [33]. Another limitation of this study is that it is designed specifically for the Australian context and may not represent the setting and conditions of other countries.

Using four high-quality general practice attributes that reflect the Quadruple Aim as a framework in this Delphi consensus study will help us to focus on the design of an assessment tool that will facilitate high-quality general practice delivery. The application of scoring criteria for approval for each consensus statement is also expected to ensure the relevance and feasibility of the final core set of indicators and measures.

Another strength of the study is the broad representation of Australian primary health organisations and diverse backgrounds of the participants involved. However, the diverse medical and non-medical participant populations with different perspectives and priorities may confound the results. If that is the case, we will be able to differentiate the stakeholder groups and analyse accordingly to identify and understand the different perspectives.

Although we have involved only PHN and general practice experts in this consensus development process, we plan to engage with primary health care consumers and Aboriginal and Torres Strait Islander health and justice health sectors separately in focus groups to explore their views on indicators and measures applicable to the final quality improvement tool. Through this Delphi consensus study, QUEST PHC will provide valuable information to guide future research and quality improvement activities in these diverse settings.

## Acknowledgments

The authors would like to acknowledge the Project Control Group: Digital Health CRC, Brisbane North PHN, Central and Eastern Sydney PHN, Nepean Blue Mountains PHN, North Western Melbourne PHN, South Western Sydney PHN, WentWest, Western Australia Primary Health Alliance, and Western NSW PHN. The authors would also like to acknowledge the RACGP, ACRRM, Justice Health NSW and SA Prison Health Service for their contribution to the Steering Committee.

## Author Contributions

**Conceptualization:** Phyllis Lau, Penelope Abbott, Kathy Tannous, Steven Trankle, Kath Peters, Andrew Page, Natalie Cochrane, Tim Usherwood, Jennifer Reath.

**Data curation:** Phyllis Lau.

**Formal analysis:** Phyllis Lau, Samantha Ryan.

**Funding acquisition:** Jennifer Reath.

**Investigation:** Phyllis Lau, Penelope Abbott, Kathy Tannous, Steven Trankle, Kath Peters, Andrew Page, Natalie Cochrane, Tim Usherwood, Jennifer Reath.

**Methodology:** Phyllis Lau, Penelope Abbott, Kathy Tannous, Steven Trankle, Kath Peters, Andrew Page, Tim Usherwood, Jennifer Reath.

**Project administration:** Phyllis Lau, Samantha Ryan.

**Resources:** Samantha Ryan.

**Software:** Samantha Ryan.

**Supervision:** Phyllis Lau.

**Writing – original draft:** Phyllis Lau.

**Writing – review & editing:** Phyllis Lau, Samantha Ryan, Penelope Abbott, Kathy Tannous, Steven Trankle, Kath Peters, Andrew Page, Natalie Cochrane, Tim Usherwood, Jennifer Reath.

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
