## [Decision Letter · Decision Letter 0]

7 Mar 2022

PONE-D-21-34478Protocol for a Delphi consensus survey to select indicators of high-quality general practice to achieve Quality Equity and Systems Transformation in Primary Health Care (QUEST-PHC) in AustraliaPLOS ONE

Dear Dr. Lau,

Thank you for submitting your manuscript to PLOS ONE. After careful consideration, we feel that it has merit but does not fully meet PLOS ONE’s publication criteria as it currently stands. Therefore, we invite you to submit a revised version of the manuscript that addresses the points raised during the review process.

We look forward to receiving your revised manuscript.

Kind regards,

Marie-Pascale Pomey

Academic Editor

PLOS ONE

Journal Requirements:

The authors would like to acknowledge the funding body and the contribution of the Project Control Group: Digital Health CRC, Brisbane North PHN, Central and Eastern Sydney PHN, Nepean Blue Mountains PHN, North Western Melbourne PHN, South Western Sydney PHN, WentWest, Western Australia Primary Health Alliance, and Western NSW PHN. 

This study is funded by the Digital Health Cooperative Research Centre https://www.digitalhealthcrc.com/. The funding body is part of the Project Control Group which oversees the conduct of the study, including the design of the study and collection, analysis, and interpretation of data and the writing the manuscript.

Reviewers' comments:

Reviewer's Responses to Questions

**Comments to the Author**

1. Does the manuscript provide a valid rationale for the proposed study, with clearly identified and justified research questions?

Reviewer #1: Yes

Reviewer #2: Yes

2. Is the protocol technically sound and planned in a manner that will lead to a meaningful outcome and allow testing the stated hypotheses?

Reviewer #1: Partly

Reviewer #2: Yes

3. Is the methodology feasible and described in sufficient detail to allow the work to be replicable?

Reviewer #1: Yes

Reviewer #2: Yes

4. Have the authors described where all data underlying the findings will be made available when the study is complete?

Reviewer #1: Yes

Reviewer #2: Yes

5. Is the manuscript presented in an intelligible fashion and written in standard English?

Reviewer #1: Yes

Reviewer #2: Yes

6. Review Comments to the Author

You may also provide optional suggestions and comments to authors that they might find helpful in planning their study.

Reviewer #1: I read this manuscript about the protocol for a Delphi study to select indicators of high-quality general practice in Australia with great interest. I would like to thank the authors for addressing this important topic and the editors for giving me the opportunity to review this protocol.

There are many strong points to this study, but I will focus my comments on issues that I believe the authors should consider to possibly improve it further. They are presented in order of occurrence in the manuscript and not of importance. Even though they are numerous, I believe most are minor and easily fixable. I was generous in my comments to reflect my interest in the methodology adopted and the chosen topic.

* Regarding point 2 - Soundness of the protocol (Minor and major points):

a) The authors mention in the abstract and main text that they will compute weighted Kappas as part of their statistical analyses. However, they provide little detail as to how these values will be used to influence the study process. Weighted kappas are normally used to assess inter-rater agreement between two raters when the data has an order structure (such as in ordinal data). The reference cited in their manuscript (#27, Holey et al. 2007) used weighted kappas to assess intra-rater (within-subject) agreement as an indication of stability of a participant's responses between two rounds. I feel this would provide little added value in the present study. Because items will only ever be assessed over a maximum of two rounds, it won't be possible to determine whether stability increases or decreases over subsequent rounds as a criteria to stop the Delphi process. Furthermore, it is expected in Delphi studies that some respondents will change their views based on the feedback provided during the second round, so I wonder how the values would be interpreted meaningfully in this case. Finally, I also have reservations about such a use of the kappa statistic because it violates its assumption of independent ratings (ratings from the same individual are expected to be correlated to a certain degree). The authors should consider removing this statistic from their analyses or better explaining how it will be used/interpreted during the research process.

b) I have not been able to access reference #11 (QUEST PHC Project Report) online. Perhaps providing more methodological details as to how the indicators and measures were produced/derived would be of interest to readers. This is briefly alluded to in the Discussion, but I think it deserves more attention either in the Introduction or in Materials and Methods.

c) I also wonder whether all of the indicators in this study are brand new or if some of them are already in use in Australia within other measurement frameworks, such as the PCMH model (there is an indirect allusion to this around lines 330-334)? If so, then surely these indicators are "feasible" in an Australian context and participants could be prevented from needlessly assessing this dimension for them?

d) The overall aim of this study, as stated in lines 137-139, mentions the development of a "professionally endorsed" tool. To me, these strong words do not resonate well with the consensus threshold (70%) that was selected. Close to a third of respondents could disagree with items that will be included in the tool. Aiming for a higher level of consensus may better convey the idea of professional endorsement. Perhaps the authors should consult with members of their target experts to determine the threshold needed to ensure the legitimacy of their tool? Or at least justify a bit more why this threshold was chosen in consideration with the context and aims of the study? See also point h) below for another issue related to the relatively weak consensus threshold adopted.

e) Regarding sample size (lines 174-178), I would like to point out that although eight participants can be enough for many Delphi studies, it is unlikely to be the case in this particular project given the aims (e.g., professional endorsement) and diversity of relevant perspectives involved in the subject matter (quality of primary care). A better reference number can be taken from a systematic review published in PLoS One, which found that the median number of panel members in Delphi studies involving the selection of healthcare quality indicators was 17 (see Boulkedid R, Abdoul H, Loustau M, Sibony O, Alberti C. Using and reporting the Delphi method for selecting healthcare quality indicators: a systematic review. PLoS One. 2011;6(6):e20476).

Also, I believe there is an error in the calculation for the minimum response rate to achieve in each round to obtain 8 respondents from an initial pool of 80 recruited (line 178): the reported rate of 32% works for a two-round process, not three. For three rounds, a minimum response rate of 46% is required in each round (although presenting the information in this way is somewhat misleading in this study's case because not all items are assessed during all rounds).

f) Around line 212: It is not specified whether and how many reminders are planned to be sent to participants to maximize response rates. This can make a big difference as we often see surges of responses shortly after sending a reminder.

g) Around lines 231-233: The authors should specify what information will be fed back to participants in subsequent rounds of rating. Feedback should not only include quantitative but also qualitative information. I have seen Delphi studies which only use quantitative feedback. In these studies, the only "reason" for participants to change their opinion is to conform to the majority, which goes against the Delphi principle of avoiding peer pressure.

h) Line 245: I believe that collapsing scores of 3 and 4 for acceptance of items further weakens the impression of consensus that will emerge from this study --- a point also related to my point d) earlier. As currently planned, half of the response scale's categories (2 out of 4) would be viewed as providing support to an item. In comparison, typical Delphi studies with response scales of 9 categories only consider a third of them (range 7-9) as support for an item. I feel that a score of 3 reflects rather mild support vs. 4 which is stronger support. By combining these categories, items with large differences in their overall level of support are likely to be included in the tool as reaching consensus. Such combination also disregards the capacity of respondents to discriminate items that is implied by the full response scale. The authors may wish to reconsider this methodological choice, e.g. by using a dual threshold that includes a minimum proportion for scores of 4 only as well as a minimum proportion for combined scores of 3 or 4. For more consideration into this issue and its consequences on the study results, see De Meyer D, Kottner J, Beele H, Schmitt J, Lange T, Van Hecke A, et al. Delphi procedure in core outcome set development: rating scale and consensus criteria determined outcome selection. J Clin Epidemiol. 2019 Jul;111:23–31.

i) Lines 251-252: Although they are frequently used in Delphi studies, means and standard deviations are generally not appropriate for ordinal data also unlikely to be normally distributed. This is especially the case with narrow response scales such as the one used in this study (wider scale can sometimes approximate interval data). Medians and interquartile ranges should be reported instead.

j) Lines 375-377: I would remove the mention that diversity of perspectives is a potential risk to achieving consensus. It should be seen as a strength, given the complex and comprehensive nature of quality in primary care.

* Regarding point 4 - Data availability (Minor points):

I am unsure whether the manuscript conforms to the PLOS Data policy. The authors wrote that data from the study planned in this protocol will be made available from the corresponding author on reasonable request and with permission of the study funder (Digital Health Cooperative Research Centre). However, PLOS Data policy states that "it is not acceptable for an author to be the sole named individual responsible for ensuring data access."

Furthermore, the authors do not specify how to obtain permission from the funder (contact information and criteria). However, I am unsure whether this is required for a protocol or only for full study reports.

* Regarding point 5 - Language considerations (Minor points):

a) I sugggest using the expression "Delphi study" or "Delphi process" rather than "Delphi survey" in the title and whenever the authors refer to the study type, since Delphi studies include multiple surveys and this could be confusing to some readers. E.g. Lines 2, 6, 198, 207.

b) Line 14: "Translational Health Research Institute" appears twice in succession.

c) Line 78: First instance of the "PHN" abbreviation should be defined.

d) Line 87: It looks like a verb is missing in this sentence? It should maybe read "The 31 PHNs [were] established in 2015 across Australia for supporting (...)"

e) Line 274: I think there is a typo in the e.g. of the survey file ("survery1" should probably read "survey").

f) I don't know if this is a misunderstanding on my part or a typo, but in Table 2, the measure "PAM scores" is presented as blue sky for the indicator O4 but not for the indicator O57. How can it be considered difficult to implement for one and not the other?

* ADDITIONAL COMMENTS FOR THE AUTHORS

Line 130: "Subsequent consultations will be held with consumers (...)". It is now increasingly common to seek involvement of consumers, patients, and communities as early as possible in the research process rather than merely at the later stages. There is evidence that professionals and patients have different priorities regarding quality improvement in primary care (e.g., see Boivin A, Lehoux P, Lacombe R, Burgers J, Grol R. Involving patients in setting priorities for healthcare improvement: a cluster randomized trial. Implement Sci. 2014 Feb 20;9:24. ). I encourage the authors to seek to involve them as early in their process as possible so that they have the opportunity to shape it just as much as professionals.

I fully understand that discussing the indicators and measures to be assessed in this study is outside the scope of my mandate as a reviewer of this protocol, but I found it unfortunate that, in Table 2, the only (two) indicators and measures for Attribute Four (Accountable to society) were labeled as blue sky. Could items related to reducing unnecessary care procedures (e.g., Choosing Wisely) have also been considered here? Would this deserve at least some discussion in the main text?

Finally, around line 336, the authors indirectly imply that the indicators/measures achieving consensus may eventually be linked to payment in reformed funding models. Are Delphi participants made aware of this? Their judgments on relevance and feasibility may differ whether they consider that the items will be used reflexively in a context of continuous quality improvement or for external sanctioning from pay-for-quality/performance schemes.

In closing, I sincerely hope that at least some of my comments will help the authors improve their manuscript and make their already good study even more robust.

Reviewer #2: This protocol paper reports the protocol for a Delphi consensus survey to select indicators of high-quality general practice to achieve Quality Equity and Systems Transformation in Primary Health Care (QUEST-PHC) in Australia.

The work is very timely and important for Australian Primary Health Care. It is also good to see outcome, as well as process, measures are being considered. Overall I strongly support the researchers' view that it "will add to the current knowledge of the translation of performance guidelines into quality practice and how best to measure and promote high quality in Australian general practice."

The protocol itself is well written and also aspects are presented appropriately. The Delphi process is an appropriate formal consensus method to use.

7. PLOS authors have the option to publish the peer review history of their article (what does this mean?). If published, this will include your full peer review and any attached files.

Reviewer #1: No

Reviewer #2: **Yes: **Professor Tim Stokes

---

## [Author Response · Author response to Decision Letter 0]

29 Mar 2022

Dear Editor,

On behalf of my co-authors, I submit below our response to the reviewers. We thank both reviewers for their time. We greatly appreciate it.

Regards,

Phyllis Lau, 15th March 2022

Additional Journal Requirements: 

AUTHORS' RESPONSE:

We have now adjusted our formatting to comply with PLOS ONE style.

The authors would like to acknowledge the funding body and the contribution of the Project Control Group: Digital Health CRC, Brisbane North PHN, Central and Eastern Sydney PHN, Nepean Blue Mountains PHN, North Western Melbourne PHN, South Western Sydney PHN, WentWest, Western Australia Primary Health Alliance, and Western NSW PHN. 

This study is funded by the Digital Health Cooperative Research Centre https://www.digitalhealthcrc.com/. The funding body is part of the Project Control Group which oversees the conduct of the study, including the design of the study and collection, analysis, and interpretation of data and the writing the manuscript.

AUTHORS' RESPONSE: 

Funding information now deleted from the Acknowledgements section. 

The reference to the funder now deleted from line 156.

Funding statement now deleted from the manuscript.

We have corrected a typo in the funding statement. A funding statement was already in the cover letter, but we have now amended to below in the cover letter so it is clearer – “This study is funded by the Digital Health Cooperative Research Centre https://www.digitalhealthcrc.com/. The funding body is part of the Project Control Group which oversees the conduct of the study, including the design of the study and collection, analysis, and interpretation of data and the writing of the manuscript.”

AUTHORS' RESPONSE: 

We have reviewed our reference list and made appropriate edits. It is now complete and correct. 

Reference 11 has been replaced – a peer reviewed paper describing findings reported in the previous reference has been accepted for publication.

Reference 23 has also been replaced with the reference suggested by reviewer 1 in point e. This reference better supports a more evidence-based recommendation of a minimum number of panel members in Delphi studies involving the selection of healthcare quality indicators.

Reviewer #1: 

I read this manuscript about the protocol for a Delphi study to select indicators of high-quality general practice in Australia with great interest. I would like to thank the authors for addressing this important topic and the editors for giving me the opportunity to review this protocol.

There are many strong points to this study, but I will focus my comments on issues that I believe the authors should consider to possibly improve it further. They are presented in order of occurrence in the manuscript and not of importance. Even though they are numerous, I believe most are minor and easily fixable. I was generous in my comments to reflect my interest in the methodology adopted and the chosen topic. 

AUTHORS' RESPONSE: 

We are grateful and thank reviewer 1 for their comments and interest in our work.

*Regarding point 2 - Soundness of the protocol (Minor and major points):

a) The authors mention in the abstract and main text that they will compute weighted Kappas as part of their statistical analyses. However, they provide little detail as to how these values will be used to influence the study process. Weighted kappas are normally used to assess inter-rater agreement between two raters when the data has an order structure (such as in ordinal data). The reference cited in their manuscript (#27, Holey et al. 2007) used weighted kappas to assess intra-rater (within-subject) agreement as an indication of stability of a participant's responses between two rounds. I feel this would provide little added value in the present study. Because items will only ever be assessed over a maximum of two rounds, it won't be possible to determine whether stability increases or decreases over subsequent rounds as a criteria to stop the Delphi process. Furthermore, it is expected in Delphi studies that some respondents will change their views based on the feedback provided during the second round, so I wonder how the values would be interpreted meaningfully in this case. Finally, I also have reservations about such a use of the kappa statistic because it violates its assumption of independent ratings (ratings from the same individual are expected to be correlated to a certain degree). The authors should consider removing this statistic from their analyses or better explaining how it will be used/interpreted during the research process. 

AUTHORS' RESPONSE: 

We have considered reviewer 1’s comment about the Kappa statistics. We agree that there are questions about the value of these statistics. We have therefore removed them from line 60 in the Abstract and line 263. 

b) I have not been able to access reference #11 (QUEST PHC Project Report) online. Perhaps providing more methodological details as to how the indicators and measures were produced/derived would be of interest to readers. This is briefly alluded to in the Discussion, but I think it deserves more attention either in the Introduction or in Materials and Methods. 

AUTHORS' RESPONSE: 

As indicated in our response to Additional Journal Requirement 3 above, we have replaced the report in reference 11 now with a recently accepted publication, currently in press. 

We have added to the methodological details (as suggested by reviewer 1) in paragraph starting line 111 – “In 2020, Western Sydney University, in partnership with PHNs in the western Sydney region, conducted a literature review to identify evidence-based indicators and measures, then assessed these in three workshops with general practitioners (GPs), practice managers, nurses, consumers and PHN staff in the western Sydney region.[11]”

c) I also wonder whether all of the indicators in this study are brand new or if some of them are already in use in Australia within other measurement frameworks, such as the PCMH model (there is an indirect allusion to this around lines 330-334)? If so, then surely these indicators are "feasible" in an Australian context and participants could be prevented from needlessly assessing this dimension for them? 

AUTHORS' RESPONSE: 

Yes, some of the indicators in the QUEST PHC suite are already currently being collected. However, our assessment of ‘feasibility’ is not assessing whether the indicators and measures are possible to use, rather we are assessing the ‘applicability and implementability’ of an indicator/measure in Australian general practice. It is also worth noting that the PCMH model has largely been studied in the USA and there has been little evaluation conducted in an Australian context. 

When considered in amongst a larger set of indicators and measures, the experts’ judgments on feasibility may differ in a context of high-quality practice and improvement. We think it is important to subject all the indicators and measures to the same assessment criteria.

d) The overall aim of this study, as stated in lines 137-139, mentions the development of a "professionally endorsed" tool. To me, these strong words do not resonate well with the consensus threshold (70%) that was selected. Close to a third of respondents could disagree with items that will be included in the tool. Aiming for a higher level of consensus may better convey the idea of professional endorsement. Perhaps the authors should consult with members of their target experts to determine the threshold needed to ensure the legitimacy of their tool? Or at least justify a bit more why this threshold was chosen in consideration with the context and aims of the study? See also point h) below for another issue related to the relatively weak consensus threshold adopted. 

AUTHORS' RESPONSE: 

Thank you for this comment. We consider the 70% threshold to be a reasonable target for establishing consensus across diverse and complex general practice settings. Furthermore, we are aiming for 70% agreement in both relevance and feasibility which is effectively consensus in two criteria, not just one.

We have added our justification to line 254 – “We determined that this threshold target and approach to be pragmatic and reasonable for establishing consensus across diverse and complex general practice settings.”

We have responded further on this issue in point h.

e) Regarding sample size (lines 174-178), I would like to point out that although eight participants can be enough for many Delphi studies, it is unlikely to be the case in this particular project given the aims (e.g., professional endorsement) and diversity of relevant perspectives involved in the subject matter (quality of primary care). A better reference number can be taken from a systematic review published in PLoS One, which found that the median number of panel members in Delphi studies involving the selection of healthcare quality indicators was 17 (see Boulkedid R, Abdoul H, Loustau M, Sibony O, Alberti C. Using and reporting the Delphi method for selecting healthcare quality indicators: a systematic review. PLoS One. 2011;6(6):e20476).

Also, I believe there is an error in the calculation for the minimum response rate to achieve in each round to obtain 8 respondents from an initial pool of 80 recruited (line 178): the reported rate of 32% works for a two-round process, not three. For three rounds, a minimum response rate of 46% is required in each round (although presenting the information in this way is somewhat misleading in this study's case because not all items are assessed during all rounds). We agree with reviewer 1 about the sample size. We have amended the sentence in line 180 – “A minimum of 17 participants is the recommended minimum sample size for content validity in Delphi studies involving the selection of healthcare quality indicators.[23]” and have also replaced ref 23 with Boulkedid et al 2011.

AUTHORS' RESPONSE: 

We thank reviewer 1 for taking the time to check our numbers. However, we believe our calculation was correct. The retaining of participants needed to occur only in rounds 2 and 3 ie only two rounds, not three. To achieve a final number of eight respondents and if we start off with 80 participants in round 1, we need to retain at least 32% (ie 25-26 participants) in round 2 and again 32% (ie 8-9 participants) in round 3.

However, since our target now is minimum of 17 participants, we will need to retain at least 47% (ie. 37-38 participants) in round 2 and and again 47% (ie. 17-18 participants) in round 3. We have therefore amended the sentence in line 182 – “In order for this Delphi study to meet the minimum sample size requirement, we must achieve a minimum of 47% retention rate in rounds 2 and 3.”

We do not agree with reviewer 1 that this is misleading as the final respondents retained in round 3 would have rated all 79 indicators with the corresponding 128 measures in the QUEST PHC suite. 

f) Around line 212: It is not specified whether and how many reminders are planned to be sent to participants to maximize response rates. This can make a big difference as we often see surges of responses shortly after sending a reminder. 

AUTHORS' RESPONSE: 

We agree with reviewer 1 and the reminder process is actually in our ethics approval. We have now added to line 219 – “Participants will receive up to three email reminders to complete each round before it closes.”

g) Around lines 231-233: The authors should specify what information will be fed back to participants in subsequent rounds of rating. Feedback should not only include quantitative but also qualitative information. I have seen Delphi studies which only use quantitative feedback. In these studies, the only "reason" for participants to change their opinion is to conform to the majority, which goes against the Delphi principle of avoiding peer pressure. 

AUTHORS' RESPONSE: 

We were very clear in our manuscript that participants will be “presented with items that did not reach consensus” in previous rounds. We do not agree that “the only reason for participants to change their opinion is to confirm to the majority” if only the quantitative ratings were presented to them. Our participants are clinicians, managers and PHN staff considered ‘experts’ in general practice quality improvement. We rely on their expert opinions in this study. For someone to review and potentially revise their opinion, informed by the opinions of others, does not imply reacting to peer pressure. The Delphi process simply provides our experts the opportunity to re-evaluate each indicator/measure while considering their peers’ responses. That is the crux of the Delphi process.

In round 3, however, we do intend to present relevant qualitative feedback from the previous rounds. We have added to line 244 – “…as the final list of indicators and measures emerges, participants will be presented with a summary of any suggestions or qualitative responses from rounds 1 and 2, and asked open questions regarding their views and suggestions on the implementation of a quality indicator tool in Australian general practice.”

h) Line 245: I believe that collapsing scores of 3 and 4 for acceptance of items further weakens the impression of consensus that will emerge from this study --- a point also related to my point d) earlier. As currently planned, half of the response scale's categories (2 out of 4) would be viewed as providing support to an item. In comparison, typical Delphi studies with response scales of 9 categories only consider a third of them (range 7-9) as support for an item. I feel that a score of 3 reflects rather mild support vs. 4 which is stronger support. By combining these categories, items with large differences in their overall level of support are likely to be included in the tool as reaching consensus. Such combination also disregards the capacity of respondents to discriminate items that is implied by the full response scale. The authors may wish to reconsider this methodological choice, e.g. by using a dual threshold that includes a minimum proportion for scores of 4 only as well as a minimum proportion for combined scores of 3 or 4. For more consideration into this issue and its consequences on the study results, see De Meyer D, Kottner J, Beele H, Schmitt J, Lange T, Van Hecke A, et al. Delphi procedure in core outcome set development: rating scale and consensus criteria determined outcome selection. J Clin Epidemiol. 2019 Jul;111:23–31. Thank you for your comment.

AUTHORS' RESPONSE: 

We have reviewed the reference suggested by reviewer 1. The De Meyer et al paper compared the inference of a 9-point and a 3-point rating scales on consensus development in one Delphi study. They concluded that “the format of rating scales in Delphi studies for core outcome set development and the definition of the consensus criteria influence outcome selection”.

When designing the Delphi study, we too considered the format of the rating scales om the context of the diverse and complex general practice settings as well as the diverse participants, including clinicians and non-clinicians, who we will be inviting. We also considered the burden of asking participants to rate 79 indicators and their corresponding 128 measures. As we alluded to in our response to point d, our approach is more pragmatic than theoretical. We consider the 70% threshold, the using of 4-point Likert scale and the combining of scores 3 and 4 to be realistic and reasonable. In our analysis, however, we also intend to differentiate the scores 1, 2, 3 and 4 to help us understand better the strength of the consensus. 

We have added this to line 270 – “Sub-analysis of the individual scores 1, 2, 3 and 4 will also be conducted to help us understand better the strength of the consensus.”

i) Lines 251-252: Although they are frequently used in Delphi studies, means and standard deviations are generally not appropriate for ordinal data also unlikely to be normally distributed. This is especially the case with narrow response scales such as the one used in this study (wider scale can sometimes approximate interval data). Medians and interquartile ranges should be reported instead. 

AUTHORS' RESPONSE: 

Thank you for your comment. We agree with you. We have deleted ‘means’ and ‘standard deviations’ from line 59 in the Abstract and line 261, and added ‘interquartile ranges’ to the ‘median’ already mentioned.

j) Lines 375-377: I would remove the mention that diversity of perspectives is a potential risk to achieving consensus. It should be seen as a strength, given the complex and comprehensive nature of quality in primary care. 

AUTHORS' RESPONSE: 

We agree with reviewer 1 that the diversity of perspectives is not a risk. We have removed the word ‘risk’ from line 387 – “However, the diverse medical and non-medical participant populations with different perspectives and priorities may confound the results.” 

Nevertheless, many of the indicators/measures are clinical, and clinicians will have very different views to non-clinicians. We will therefore retain the sentence following in line 389 that states our intention to conduct sub-analysis to help us understand the differences, if any.

* Regarding point 4 - Data availability (Minor points):

I am unsure whether the manuscript conforms to the PLOS Data policy. The authors wrote that data from the study planned in this protocol will be made available from the corresponding author on reasonable request and with permission of the study funder (Digital Health Cooperative Research Centre). However, PLOS Data policy states that "it is not acceptable for an author to be the sole named individual responsible for ensuring data access."

Furthermore, the authors do not specify how to obtain permission from the funder (contact information and criteria). However, I am unsure whether this is required for a protocol or only for full study reports. 

AUTHORS' RESPONSE: 

The corresponding author will not be the “sole named individual responsible for ensuring data access”. QUEST PHC has a project agreement with the Digital Health Cooperative Research Centre (DH CRC). The project IP will be owned by DHCRC legally; however, the authors will have non-exclusive royalty-free right to use the DHCRC IP for purposes including publication and conference presentations.

The role of the corresponding author, we thought, were to be a point of contact for the journal and any questions or requests from readers. Provided the requests for data do not compromise confidential information, are consistent with the purpose of the funded project, and do not prejudice the protection or utilisation of the DH CRC IP, they will be honoured in accordance with our project agreement with DH CRC. 

* Regarding point 5 - Language considerations (Minor points):

a) I sugggest using the expression "Delphi study" or "Delphi process" rather than "Delphi survey" in the title and whenever the authors refer to the study type, since Delphi studies include multiple surveys and this could be confusing to some readers. E.g. Lines 2, 6, 198, 207. 

AUTHORS' RESPONSE: 

Amended references from ‘Delphi survey’ to ‘Delphi study’ wherever appropriate throughout the manuscript.

Please note that in line 213, we have retained the word ‘survey’ as the word is appropriate. 

b) Line 14: "Translational Health Research Institute" appears twice in succession. 

AUTHORS' RESPONSE: 

Deleted repetition.

c) Line 78: First instance of the "PHN" abbreviation should be defined. 

AUTHORS' RESPONSE: 

‘PHN’ expanded.

d) Line 87: It looks like a verb is missing in this sentence? It should maybe read "The 31 PHNs [were] established in 2015 across Australia for supporting (...)" 

AUTHORS' RESPONSE: 

Amended as suggested.

e) Line 274: I think there is a typo in the e.g. of the survey file ("survery1" should probably read "survey"). 

AUTHORS' RESPONSE: 

It was not a typo. This is the nomenclature we are using to name our files.

f) I don't know if this is a misunderstanding on my part or a typo, but in Table 2, the measure "PAM scores" is presented as blue sky for the indicator O4 but not for the indicator O57. How can it be considered difficult to implement for one and not the other? 

AUTHORS' RESPONSE: 

Thank you to reviewer 1 for the pickup. It is an error – the PAM scores should not be ‘blue sky’. This is now corrected in Table 2. 

This manuscript has taken a long time to be reviewed, and data collection for the survey is already underway at the time of writing this response. We will note this error to participants in the final Round 3.

* ADDITIONAL COMMENTS FOR THE AUTHORS

Line 130: "Subsequent consultations will be held with consumers (...)". It is now increasingly common to seek involvement of consumers, patients, and communities as early as possible in the research process rather than merely at the later stages. There is evidence that professionals and patients have different priorities regarding quality improvement in primary care (e.g., see Boivin A, Lehoux P, Lacombe R, Burgers J, Grol R. Involving patients in setting priorities for healthcare improvement: a cluster randomized trial. Implement Sci. 2014 Feb 20;9:24. ). I encourage the authors to seek to involve them as early in their process as possible so that they have the opportunity to shape it just as much as professionals.

AUTHORS' RESPONSE: 

We thank reviewer 1 for this comment. We agree that early engagement of consumers is very important. The QUEST PHC suite is a broad set of indicators that include both clinical and patient measures. We engaged with consumers at the outset of this work and they advised that whilst they understood and supported the need for clinical measures, they were most interested to be involved in the development of Patient reported measures. We have recently completed a literature review to identify patient-reported measures (PRMs) that are relevant to Australian general practice. At the time of writing this response, we have already conducted focus groups with consumers to explore their views on PRMs. 

The sentence starting in line 135 has now been amended to “Consultations have been held with consumers, with regards to key patient-reported measures (PRMs).”

I fully understand that discussing the indicators and measures to be assessed in this study is outside the scope of my mandate as a reviewer of this protocol, but I found it unfortunate that, in Table 2, the only (two) indicators and measures for Attribute Four (Accountable to society) were labeled as blue sky. Could items related to reducing unnecessary care procedures (e.g., Choosing Wisely) have also been considered here? Would this deserve at least some discussion in the main text? 

AUTHORS' RESPONSE: 

The development of the indicators and measures were informed by extensive literature reviews and consultations with primary health networks and clinicians. We accept that not all important indicators are considered ‘practical’ by all stakeholders. The use of the ‘blue sky’ label is precisely to preserve such indicators until such time the environment is right for their use. 

We have amended the sentence that mentions this in line 127 to be clearer – “…include some “blue sky” measures considered difficult to currently implement but are nonetheless important.”

Finally, around line 336, the authors indirectly imply that the indicators/measures achieving consensus may eventually be linked to payment in reformed funding models. Are Delphi participants made aware of this? Their judgments on relevance and feasibility may differ whether they consider that the items will be used reflexively in a context of continuous quality improvement or for external sanctioning from pay-for-quality/performance schemes. 

AUTHORS' RESPONSE: 

Yes, participants are made aware. 

Our Participant Information Sheet clearly states that “This research will provide support for use of these measures to improve care in general practices and potentially inform funding models that would reward high quality care in Australian general practice.”. 

In closing, I sincerely hope that at least some of my comments will help the authors improve their manuscript and make their already good study even more robust. 

AUTHORS' RESPONSE: 

We thank reviewer 1 and have no doubt that their very considered review will make our study more robust. 

Reviewer #2: 

This protocol paper reports the protocol for a Delphi consensus survey to select indicators of high-quality general practice to achieve Quality Equity and Systems Transformation in Primary Health Care (QUEST-PHC) in Australia.

The work is very timely and important for Australian Primary Health Care. It is also good to see outcome, as well as process, measures are being considered. Overall I strongly support the researchers' view that it "will add to the current knowledge of the translation of performance guidelines into quality practice and how best to measure and promote high quality in Australian general practice."

The protocol itself is well written and also aspects are presented appropriately. The Delphi process is an appropriate formal consensus method to use. 

AUTHORS' RESPONSE: 

We thank reviewer 2 for their encouraging and generous comments!

---

## [Editor Report · Decision Letter 1]

22 Apr 2022

Protocol for a Delphi consensus study to select indicators of high-quality general practice to achieve Quality Equity and Systems Transformation in Primary Health Care (QUEST-PHC) in Australia

PONE-D-21-34478R1

Dear Dr. Min-yu Lau,

We’re pleased to inform you that your manuscript has been judged scientifically suitable for publication and will be formally accepted for publication once it meets all outstanding technical requirements.

Kind regards,

Marie-Pascale Pomey

Academic Editor

PLOS ONE

---

## [Editor Report · Acceptance letter]

12 May 2022

PONE-D-21-34478R1 

Protocol for a Delphi consensus study to select indicators of high-quality general practice to achieve Quality Equity and Systems Transformation in Primary Health Care (QUEST-PHC) in Australia 

Dear Dr. Lau:

I'm pleased to inform you that your manuscript has been deemed suitable for publication in PLOS ONE. Congratulations! Your manuscript is now with our production department. 

Kind regards, 

on behalf of

Dr. Marie-Pascale Pomey 

Academic Editor

PLOS ONE